# CodeXGLUE: A Machine Learning Benchmark Dataset for Code Understanding and Generation

**Shuai Lu**[*]
Peking University

**Daya Guo**[*]
Sun Yat-sen University

**Shuo Ren**[*]
Beihang University

**Junjie Huang**[*]
Beihang University

**Alexey Svyatkovskiy**
Microsoft

**Ambrosio Blanco**
Microsoft Research Asia

**Colin Clement**
Microsoft

**Dawn Drain**
Microsoft

**Daxin Jiang**
Microsoft

**Duyu Tang**
Microsoft Research Asia

**Ge Li**
Peking University

**Lidong Zhou**
Microsoft Research Asia

**Linjun Shou**
Microsoft

**Long Zhou**
Microsoft Research Asia

**Michele Tufano**
Microsoft

**Ming Gong**
Microsoft

**Ming Zhou**
Microsoft Research Asia

**Nan Duan**
Microsoft Research Asia

**Neel Sundaresan**
Microsoft

**Shao Kun Deng**
Microsoft

**Shengyu Fu**
Microsoft

**Shujie Liu**
Microsoft Research Asia

## Abstract

Benchmark datasets have a significant impact on accelerating research in programming language tasks. In this paper, we introduce CodeXGLUE, a benchmark dataset to foster machine learning research for program understanding and generation. CodeXGLUE includes a collection of 10 tasks across 14 datasets and a platform for model evaluation and comparison. CodeXGLUE also features three baseline systems, including the BERT-style, GPT-style, and Encoder-Decoder models, to make it easy for researchers to use the platform. The availability of such data and baselines can help the development and validation of new methods that can be applied to various program understanding and generation problems. [2]

## 1 Introduction

Evans Data Corporation[3] estimated that there were 23.9 million professional developers in 2019 and that the number was expected to reach 28.7 million in 2024. With the population of developers growing at such a rate, code intelligence that leverages artificial intelligence (AI) to help software developers improve the productivity of the development process is becoming increasingly important. It is commonly accepted that benchmarks have a significant impact on the growth of applied AI research. In this paper, we focus on establishing a benchmark dataset for code intelligence.

---

[*]indicates equal contribution and internship at Microsoft. Authors are listed in alphabetical order. Corresponding author is Nan Duan.

[2]CodeXGLUE is publicly available at `https://github.com/microsoft/CodeXGLUE`. Participants can submit their results by emailing to `codexglue@microsoft.com`.

[3]`https://evansdata.com/press/viewRelease.php?pressID=278`

35th Conference on Neural Information Processing Systems (NeurIPS 2021) Track on Datasets and Benchmarks.

Table 1: A brief summary of CodeXGLUE, which includes tasks, datasets, languages, sizes in various states, and baseline systems. [†] indicates newly introduced datasets and [‡] means we introduce a new part for the existing dataset.

| Category | Task | Dataset Name | Language | Train/Dev/Test Size | Baselines |
|---|---|---|---|---|---|
| Code-Code | Clone Detection | BigCloneBench [65] | Java | 900K/416K/416K | CodeBERT |
| | | POJ-104 [46] | C/C++ | 32K/8K/12K | |
| | Defect Detection | Devign [91] | C | 21K/2.7K/2.7K | |
| | Cloze Test | CT-all[†] | Python,Java,PHP, JavaScript,Ruby,Go | -/-/176K | |
| | | CT-max/min[†] [15] | Python,Java,PHP, JavaScript,Ruby,Go | -/-/2.6K | |
| | Code Completion | PY150 [57][‡] | Python | 95K/5K/50K | CodeGPT |
| | | Github Java Corpus[1][‡] | Java | 13K/7K/8K | |
| | Code Repair | Bugs2Fix [69] | Java | 98K/12K/12K | Encoder-Decoder |
| | Code Translation | CodeTrans[†] | Java-C# | 10K/0.5K/1K | |
| Text-Code | NL Code Search | CodeSearchNet [32], AdvTest[†] | Python | 251K/9.6K/19K | CodeBERT |
| | | CodeSearchNet [32], WebQueryTest[†] | Python | 251K/9.6K/1K | |
| | Text-to-Code Generation | CONCODE [34] | Java | 100K/2K/2K | CodeGPT |
| Code-Text | Code Summarization | CodeSearchNet [32] | Python,Java,PHP, JavaScript,Ruby,Go | 908K/45K/53K | Encoder-Decoder |
| Text-Text | Documentation Translation | Microsoft Docs[†] | English-Latvian/Danish /Norwegian/Chinese | 156K/4K/4K | |

Automated program understanding and generation could increase the productivity of software developers. In fact, developers who want to find code written by others with the same intent can leverage code search systems [32, 20, 81, 52] to automatically retrieve semantically relevant codes through natural language queries. Similarly, developers who are confused about what to write next can use code completion systems [57, 1, 56, 66, 67, 8, 27, 7] to automatically complete the following tokens based on the edits made to the code.

In recent years, researchers have increasingly applied statistical models, including neural networks, to code intelligence tasks. Very recently, the application of pretrained models that learn from big programming language corpus has been inspired by the great success of pretrained models like BERT [14] and GPT [63] in natural language processing (NLP). These models, including CodeBERT [15] and IntelliCode Compose [67], have led to further improvements in code understanding and generation problems. However, the area of code intelligence lacks a benchmark suite that covers a wide range of tasks. The use of ImageNet [13] for computer vision and the use of GLUE [76] for NLP have shown that a diversified benchmark dataset has a significant impact on the growth of applied AI research.

To address this problem, we introduce CodeXGLUE, a machine learning benchmark dataset for program understanding and generation research that includes 14 datasets, a collection of 10 diversified programming language understanding and generation tasks,[4] and a platform for model evaluation and comparison. CodeXGLUE supports the following tasks:

- **code-code** (clone detection [65, 84, 46, 80, 9, 89, 86], defect detection [91, 55, 51, 42, 78, 79], cloze test [15], code completion [57, 1, 56, 66, 67, 8, 27, 7], code repair [69, 4, 23, 26, 71, 73], and code-to-code translation [48, 37, 10, 41])

- **text-code** (natural language code search [32, 20, 81], text-to-code generation [34, 11, 87, 83, 85, 88, 35, 22])

- **code-text** (code summarization [33, 11, 16, 75, 81, 83, 3, 30, 82])

- **text-text** (documentation translation [36])

CodeXGLUE includes eight previously proposed datasets — BigCloneBench [65], POJ-104 [46], Devign [91], PY150 [57], Github Java Corpus [1], Bugs2Fix [69], CONCODE [34], and CodeSearchNet [32]— but also newly introduced datasets that are marked in Table 1. The datasets are chosen or created based on the consideration that the task has clear definition, the popularity and reputation

---

[4]We plan to evolve the benchmark over time by extending to more tasks.

of the dataset in the community, and the volume of the dataset could support the development and evaluation of data-driven machine learning methods. The datasets created by us include (1) two test sets for cloze test that cover 6 programming languages, (2) two line-level code completion test sets in Java and Python, respectively, (3) a code-to-code translation dataset between Java and C#, (4) two natural language code search test sets with web queries and normalized function and variable names, respectively, and (5) a documentation translation dataset that covers five natural languages.

To make it easy for participants, we provide three baseline models to help perform the tasks, including a BERT-style pretrained model (in this case, CodeBERT) to supports code understanding problems, a GPT-style pretrained model, which we call CodeGPT, to help solve completion and generation problems, and an Encoder-Decoder framework that tackles sequence-to-sequence generation problems.

## 2 Tasks and Datasets

CodeXGLUE consists of 14 different datasets falling into 10 diversified tasks. In this section, we provide a definition for each task and describe the dataset details. All the datasets are available under a permissive license that allows computational use purposes, such as artificial intelligence, machine learning, and text and data mining.

**Tasks Choice** CodeXGLUE aims to covering the most common scenarios for software development. Assuming there is a developer who wants to build a software system. Since it is difficult to build from scratch, he would like to start with an open-source project to get familiar with how to build a software system. But the developer still finds it difficult to understand the intention of some methods written by others because no documentation is provided. So he needs code summarization systems to generate comments for source codes. In another case, if the documentation is written in a language he can't understand, he can use documentation translation systems to translate the comments. When the developer starts writing codes, he might be confused about what to write next. Code recommendation systems can give him suggestions on the next token, the unfinished line or even a whole method and improve the efficiency of the development process, which is the code completion task and code generation task's purpose. From time to time, he would be stuck by implementing a specific function, e.g., the quick sort algorithm. What he is most likely to do next is to search "how to implement quick sort algorithm" in a search engine and expect to find a solution. This is what the natural language code search task aims to doing. When the developer finds a code snippet which could be helpful to him but he wants to implement the same function in another programming language, he can leverage code translation systems to translate one function from one programming language to another. When he finally finishes building a software system, there might be some bugs or defects in his codes. The defect detection task and the code repair task are to find the bugs and defects and auto-fix them. During the process described above, the developer reuses code fragments from other projects, which leads the software system containing clone codes. Studies [6, 60] have shown that a software system usually contain 7-23% clone codes. Detecting and refactoring cloned codes are beneficial for maintaining the software quality. That's why we add the clone detection task. And the cloze test task is served as a probing task, which has been widely used in NLP [50, 68]. It can intuitively evaluate models' ability to understand code semantics.

### 2.1 Clone detection

Clone detection is to measure the semantic similarity between source codes. It includes two subtasks. The first subtask is to predict whether two given codes have the same semantics. We use the BigCloneBench [65] dataset for the subtask. The second subtask aims to retrieve semantically similar codes given a code as the query and we use the dataset POJ-104 [46] to perform it.

**BigCloneBench** [65] is a widely used large code clone benchmark that contains over 6,000,000 true clone pairs and 260,000 false clone pairs from 10 different functionalities in 25,000 projects. Each code fragment is a Java method and all possible pairwise combinations of code fragments are clone pairs. We follow Wang et al. [80], filtering the dataset by discarding code fragments that are not in any tagged true or false clone pairs, leaving it with 9,134 Java code fragments. Finally, the dataset includes 901,028/415,416/415,416 clone pairs for training, validation and testing, respectively. Since the number of false clone pairs is much more than that of true clone pairs, we use F1 score as the metric.

**POJ-104** [46] comes from a pedagogical programming open judge (OJ) system that automatically judges the validity of submitted source code for specific problems by running the code. POJ-104 consists of 104 problems and each problem includes 500 student-written C/C++ programs. Different from that of the BigCloneBench dataset, the task of POJ-104 aims to retrieve other 499 programs that solve the same problem given a program. We group the dataset in three subsets based on the number of problems (64/16/24) for training, validation, and testing. Mean Average Precision (MAP) is used as the metric.

## 2.2 Defect detection

The task is formulated as a binary classification to predict whether a code snippet contains defects. We use **Devign** which is provided by Zhou et al. [91] and includes 27,318 manually-labeled functions collected from two large C programming language open-source projects popular among developers and diversified in functionality, i.e., QEMU and FFmpeg. The dataset is created by collecting security-related commits and extracting vulnerable or non-vulnerable functions from the labeled commits. Since Zhou et al. [91] do not provide official training/validation/testing split, we randomly shuffle the dataset and split it by 8:1:1 for training/validation/testing. Accuracy is used as the metric.

## 2.3 Cloze test

The cloze test (CT) task in code domain aims to assess models' ability to understand a code by asking those models to predict the masked code from several candidates. We focus on two subtasks: CT-all with candidates from a filtered vocabulary and CT-maxmin with the candidates "max" and "min". Accuracy is used as the metric for both datasets.

We use the validation and testing sets of CodeSearchNet [32] to create CT-all and CT-maxmin datasets for six programming languages, i.e., Go, Java, JavaScript (JS), PHP, Python and Ruby. The data statistics are listed in Table 2.

**CT-all** is created by masking the cloze words in source codes. To less introduce lengthy variable names and avoid the issue caused by the use of different tokenizers, we select target cloze words by retaining unique words after Byte Pair Encoding [61], and we remove meaningless tokens like punctuations with handcrafted rules. At last, 930 tokens are selected among six languages in total. We select codes containing the 930 tokens and manually set threshold values of token occurrence to balance the frequency of the 930 tokens in CT-all.

To further evaluate models' ability to understand code semantics, we introduce **CT-maxmin** to test how well model can distinguish the difference between *max* and *min*. It comes from the dataset used for the PL-Probing task in CodeBERT[15].

Table 2: Data statistics about the cloze test datasets.

| Task | Go | Java | JavaScript | PHP | Python | Ruby | All |
|------|------|------|------------|------|--------|------|--------|
| CT-all | 25,282 | 40,492 | 13,837 | 51,930 | 40,137 | 4,437 | 176,115 |
| CT-maxmin | 152 | 482 | 272 | 407 | 1,264 | 38 | 2,615 |

## 2.4 Code completion

Code completion task aims to predict the following tokens based on the code context. Its subtasks are token-level completion and line-level completion. The former is to predict the next one token, while the latter requires models to complete an unfinished line. We use two influential datasets, **PY150**[57] in python and **Github Java Corpus**[1] in Java. Accuracy is used as the metric for token-level completion. Exact match and Levenshtein edit similarity are served as metrics in the line-level completion subtask.

**PY150** [57] is a Python dataset containing 150,000 Python source files collected from Github. We use the same 50,000 files for testing as the original split and 95,000/5,000 files for training and validation. We preprocess the corpora by tokenizing source codes, normalizing uncommon literals as introduced by Svyatkovskiy et al. [67], and adding a special token $\langle EOL \rangle$ (end-of-line) to mark the ending of a line explicitly. For line-level code completion, we create 10,000 examples from different files in

the test set of PY150 for testing. Since we intend to test model's ability to autocomplete an arbitrary line, we select the line to be predicted at random. We generate a test case by ensuring that there is sufficient context, i.e., at least 15% of the whole file.

**Github Java Corpus** is a Java dataset mined by Allamanis and Sutton [1], and it collects over 14 thousand Java projects from Github. We follow the settings established by Hellendoorn and Devanbu [25] , using 1% of the subset in the corpus. We have 12,934/7,189/8,268 files for training/validation/testing. We do the same preprocessing conducted on PY150, except adding the special token $\langle EOL \rangle$ since in Java, semicolons and closing braces could indicate the ending of a code statement. For line-level code completion, we create 3,000 examples from different files in the test set of the corpus.

## 2.5 Code repair

Code repair aims to fix bugs in the code automatically. We use the **Bugs2Fix** dataset released by Tufano et al. [69]. The source is buggy Java functions, whereas the target is the corresponding fixed functions. To build this dataset, they collect every public GitHub event between March 2011 and October 2017 and identify all Java-file commits having a message containing the patterns [18]: ("fix" or "solve") and ("bug" or "issue" or "problem" or "error"). For each bug-fixing commit, they extract the source code before and after the fixing process. Subsequently, they normalize all the names of the variables and custom methods, which enables the model to focus on learning bug-fixing patterns. They filter out the pairs that contain lexical or syntactic errors in either the buggy or fixed code, as well as the pairs with more than 100 atomic AST modification actions between the buggy and the fixed versions. Finally, they divide the whole dataset into two subsets based on the code length(*small* with tokens $\leq 50$ and *medium* with tokens $> 50$ and $\leq 100$). For the *small* subset, the numbers of training, validation, and testing samples are 46,680, 5,835, and 5,835, respectively. For the *medium* subset, the numbers are 52,364, 6,545, and 6,545, respectively. Exact match accuracy, BLEU and CodeBLEU are used as the metrics.

## 2.6 Code translation

This task involves translating a code snippet from one programming language to a different one. In this paper, we provide **CodeTrans** which consists of parallel code snippets between Java and C#. Following Nguyen et al. [48] and Chen et al. [10], we collect source codes from several open-source projects, i.e., Lucene[5], POI[6], JGit[7] and Antlr[8]. Those projects are originally developed in Java and then ported to C#. They are well-established systems with long developing histories and with both Java and C# versions in use.

Following Nguyen et al. [48], we conservatively search for the functions having the same signatures in the classes with the same/similar names and included in the same/similar directory structures of both Java and C# versions. We discard duplicate code pairs and the codes having multiple targets searched with the above method. We also remove the pairs whose number of overlapping tokens was less than 1/3 of the sentence length. To make our data more scalable for further syntactic and semantic analysis, we remove the functions with null function body according to their abstract syntax tree (AST). Finally, a function with no data-flow extracted from the AST of a specific function is also discarded.

At last, the total number of paired functions or methods is 11,800. We randomly select 500/1,000 pairs for validation/testing, leaving 10,300 pairs for training. Exact match accuracy, BLEU and CodeBLEU[59] are used as the metrics.

## 2.7 Code search

Code search measures the semantic relatedness between texts and codes. It includes two subtasks. The first one is to find the most relevant code from a collection of candidates given a natural language query. We create a challenging testing set, called **CodeSearchNet AdvTest**, from CodeSearchNet

---

[5]http://lucene.apache.org/

[6]http://poi.apache.org/

[7]https://github.com/eclipse/jgit/

[8]https://github.com/antlr/

corpus [32] for performing this task. The second subtask is a binary classification to predict whether a code answers a given query. We provide **WebQueryTest** of real user queries.

**CodeSearchNet AdvTest** is the abbreviation of CodeSearchNet Adversarial Test Dataset. It is a Python dataset from the CodeSearchNet [32] corpus. Each example includes a function paired with a document. Following Husain et al. [32] to take the first paragraph of the documentation as the query for the corresponding function, we obtain a dataset with 251,820/9,604/19,210 examples for training/validation/testing after filtering some low-quality examples (See Appendix A). Different from CodeSearchNet[32], to better test models' understanding and generalization abilities, we normalize function and variable names in validation and testing sets like $func$ for the function name and $arg_i$ for the i-th variable name, which makes **CodeSearchNet AdvTest** dataset more difficult. In contrast to the testing phase of previous works [32, 15] that only involved 1,000 candidates, we use the entire testing set as candidates for each query. Mean Reciprocal Rank (MRR) is used as the metric for this dataset.

**WebQueryTest**. Most code search datasets use code documentations or questions from online communities for software developers as queries, but these are different from real user search queries. To fix this discrepancy, we provide WebQueryTest, a testing set of real code search for Python. The problem is formulated as a binary classification task and as a complementary setting to the retrieval scenario. Given a pair of query and code function, a model needs to classify whether the code function can answer the query or not. We invite 13 developers proficient in Python to annotate the examples and finally collect 1,046 labels of query and code pairs. The details of data collection and annotation are available in Appendix B. Since there lacks a direct training and validation set, we use the CoSQA [31] as the training resources with 20,604 pairs of query and code. F1 score is used as the metric.

## 2.8 Text-to-code generation

Text-to-code generation aims to generate source code via a natural language description. To carry out this task, we use **CONCODE** [34], a widely used code generation dataset, which is collected from about 33,000 Java projects on GitHub. It contains 100,000 examples for training and 2,000 examples each for validation and testing. Each example is a tuple consisting of NL descriptions, code environments and code snippets. The dataset is tasked with generating class member functions from natural language descriptions (Javadoc-style method comments) and class environments. Class environment is the programmatic context provided by the rest of the class, including other member variables and member functions in the class. Exact match accuracy, BLEU and CodeBLEU are used as the metrics.

## 2.9 Code summarization

The objective is to generate the natural language comment for a code. We use the **CodeSearchNet** dataset [32] for this task. We take the first paragraph as the documentation and filter low-quality examples as we do in Section 2.7. The statistics about the filtered CodeSearchNet dataset are listed in Table 3. Exact match accuracy, BLEU and CodeBLEU are used as the metrics.

Table 3: Data statistics about the filtered CodeSearchNet dataset for the code summarization task.

| Data Split | Go | Java | JavaScript | PHP | Python | Ruby |
|---|---|---|---|---|---|---|
| Training | 167,288 | 164,923 | 58,025 | 241,241 | 251,820 | 24,927 |
| Validation | 7,325 | 5,183 | 3,885 | 12,982 | 13,914 | 1,400 |
| Testing | 8,122 | 10,955 | 3,291 | 14,014 | 14,918 | 1,261 |

## 2.10 Documentation translation

Documentation translation aims to translate code documentations automatically from one natural language (e.g., English) to another natural language (e.g., Chinese). The dataset we use in CodeXGLUE is crawled from Microsoft Documentation[9], including software and code description documentations in different languages. We introduce multilingual machine translation tasks, e.g., English ⇔ Latvian, Danish, Norwegian, and Chinese. We filter the corpus (See Appendix A) to improve the data quality.

---

[9]https://docs.microsoft.com, whose document is located at https://github.com/MicrosoftDocs/.

The final training data includes 43K, 19K, 44K, and 50K sentence pairs for English ⇔ Latvian, English ⇔ Danish, English ⇔ Norwegian, and English ⇔ Chinese, respectively. In addition, each language pair has 1K sentence pairs for validation and testing, respectively. BLEU is used as the metric.

## 3 Baseline Systems

We provide three types of baseline models to perform the previously mentioned tasks, including a BERT-style pretrained model (in this case, CodeBERT[15]), which supports program understanding problems, a GPT-style pretrained model called CodeGPT that helps us solve completion and generation problems, and an Encoder-Decoder framework that tackles sequence-to-sequence generation problems.

### 3.1 CodeBERT

To carry out code understanding tasks like clone detection, defect detection, cloze test, and code search, we use CodeBERT [15] as our encoder. This is a bimodal pretrained model based on Transformer with 12 layers for programming language (PL) and natural language (NL). Feng et al. [15] introduce CodeBERT, which is initialized with RoBERTa and further trained by masked language modeling and replaced token detection objectives on the CodeSearchNet dataset [32], which includes 2.4M functions with document pairs for six programming languages. The model is publicly available at `https://huggingface.co/microsoft/codebert-base`.

### 3.2 CodeGPT

We provide CodeGPT, which is a Transformer-based language model pretrained on programming language (PL), to support the code completion and the text-to-code generation tasks. CodeGPT has the same model architecture and training objectives of GPT-2 [53], which consists of 12 layers of Transformer decoders.

We train CodeGPT and CodeGPT-adapted. CodeGPT is pretrained from scratch on Python and Java corpora from the CodeSearchNet dataset [32], which includes 1.1M Python functions and 1.6M Java methods. CodeGPT-adapted is further trained on CodeSearchNet from GPT-2 checkpoint. Both models are publicly available at `https://huggingface.co/microsoft/CodeGPT-small-java` and `https://huggingface.co/microsoft/CodeGPT-small-java-adaptedGPT2`. [10]

### 3.3 Encoder-Decoder

For sequence-to-sequence generation problems like code repair, code translation, code summarization, and documentation translation, we provide an Encoder-Decoder framework. We initialize the encoder using CodeBERT [15] and use a randomly initialized Transformer with 6 layers, 768 dimensional hidden states and 12 attention heads as the decoder in all settings. We refer to this model as **CodeBERT-EncDec**.

## 4 Experiment

In this section, we report accuracy numbers of the baseline systems on 10 tasks.

**Clone Detection**    Results achieved by different models are shown in Table 4. **ASTNN** [89] uses RNNs to encode AST subtrees for statements. **FA-AST-GMN** [80] uses GNNs over a flow-augmented AST to leverage explicit control and data flow information. **Aroma** [45] is a code recommendation engine that takes a partial code snippet and recommends a small set of succinct code snippets that contain the query snippet. **MISIM-GNN** [86] learns a structural representation of code from context-aware semantic structure designed specifically to lift semantic meaning from the code syntax. **RoBERTa** [44] and **CodeBERT** [15] are pretrained models which encode source code and take the representation to calculate semantic relevance of two code snippets through a feed forward network or inner product. These experimental results demonstrate that pretrained models are comparable

---

[10]Replace "java" with "py" for models pretrained on python dataset.

with previous state-of-the-art models. And there is room for further improvement if code structure is further leveraged like in other AST-based models.

Table 4: Results on the clone detection task.

| | BigCloneBench | POJ-104 | |
| --- | --- | --- | --- |
| Model | F1 | MAP | Overall |
| ASTNN | 93.0 | - | - |
| FA-AST-GMN | **95.0** | - | - |
| Aroma | - | 55.12 | - |
| MISIM-GNN | - | 82.45 | - |
| RoBERTa | 91.3 | 76.67 | 84.0 |
| CodeBERT | 94.1 | **82.67** | **88.4** |

**Defect Detection**    Table 5 shows the results of the models we implemented. We use Bidirectional LTSM (**BiLTSM**) [28], **TextCNN** [38], **RoBERTa** [44], and **CodeBERT** [15] to encode the representation of a source code, respectively. Then, a two-layer feed forward network followed by a softmax layer is used to calculate the probability of encountering vulnerabilities.

Table 5: Results on the defect detection task.

| Model | Accuracy |
| --- | --- |
| BiLSTM | 59.37 |
| TextCNN | 60.69 |
| RoBERTa | 61.05 |
| CodeBERT | **62.08** |

**Cloze test**    Table 6 shows the results on the CT-all and CT-maxmin datasets. We report the performance of **RoBERTa** [44] and **CodeBERT (MLM)** [15], which is pretrained with the masked language modeling objective only.

Table 6: Results on the cloze test task.

| Model | CT-all | CT-maxmin | Overall |
| --- | --- | --- | --- |
| | Ruby / JS / Go / Python / Java / PHP | | |
| RoBERTa | 47.44 / 59.96 / 40.77 / 54.35 / 50.73 / 60.16 | 73.68 / 64.71 / 71.71 / 59.18 / 59.75 / 69.78 | 59.35 |
| CodeBERT(MLM) | **80.17 / 81.77 / 83.31 / 87.21 / 80.63 / 85.05** | **86.84 / 86.40 / 90.79 / 82.20 / 90.46 / 88.21** | **85.25** |

**Code completion**    Table 7 shows the results of all models on both datasets. We train **LSTM** [28], **Transformer** [72] and fine-tune **GPT-2** [53], **CodeGPT** and **CodeGPT-adapted** to generate the following tokens. The overall score on each dataset is the average value of the accuracy on token-level completion and the edit similarity on line-level completion.

**Code repair**    Results are shown in Table 8. The **Naïve** method directly copies the buggy code as the repair result. With regard to the **CodeBERT-EncDec** method, we use the training data to fine-tune the whole model. The CodeBLEU score is used as the overall score.

**Code translation**    Table 9 shows the results of models on both translation directions. The **Naïve** method directly copies the source code as the translation result. **PBSMT** is short for phrase-based statistical machine translation [39]. **RoBERTa (code)** is initialized by RoBERTa and pretrained on source code from CodeSearchNet[32]. The overall score is the average value of CodeBLEU on both directions.

Table 9: Results on the code translation task.

| Model | Java→C# | | | C#→Java | | | Overall |
| --- | --- | --- | --- | --- | --- | --- | --- |
| | BLEU | Acc | CodeBLEU | BLEU | Acc | CodeBLEU | |
| Naïve | 18.54 | 0.0 | - | 18.69 | 0.0 | - | - |
| PBSMT | 43.53 | 12.5 | 42.71 | 40.06 | 16.1 | 43.48 | 43.10 |
| RoBERTa (code) | 77.46 | 0.561 | 83.07 | 71.99 | 0.579 | **80.18** | 81.63 |
| CodeBERT-EncDec | **79.92** | **59.0** | **85.10** | 72.14 | **58.0** | 79.41 | **82.26** |

Table 7: Results on the code completion task.

| Model | PY150 | | | Github Java Corpus | | | Overall |
|---|---|---|---|---|---|---|---|
| | token-level | line-level | | token-level | line-level | | |
| | Accuracy | EM | Edit Sim | Accuracy | EM | Edit Sim | |
| LSTM | 61.94 | 23.77 | 56.26 | 58.92 | 12.97 | 42.10 | 54.81 |
| Transformer | 74.48 | 38.51 | 69.01 | 65.18 | 17.00 | 50.23 | 64.73 |
| GPT-2 | 75.90 | 41.73 | 70.60 | 75.40 | 27.50 | 60.36 | 70.57 |
| CodeGPT | 76.58 | 42.18 | 71.23 | 76.79 | 28.23 | 61.81 | 71.60 |
| CodeGPT-adapted | **76.60** | **42.37** | **71.59** | **77.73** | **30.60** | **63.45** | **72.34** |

Table 8: Results on the code repair task.

| Model | small | | | medium | | |
|---|---|---|---|---|---|---|
| | BLEU | Acc | CodeBLEU | BLEU | Acc | CodeBLEU |
| Naïve | **78.06** | 0.0 | - | 90.91 | 0.0 | - |
| LSTM | 76.76 | 10.0 | - | 72.08 | 2.5 | - |
| Transformer | 77.21 | 14.7 | 73.31 | 89.25 | 3.7 | 81.72 |
| CodeBERT-EncDec | 77.42 | **16.4** | **75.58** | **91.07** | **5.2** | **87.52** |

**Code search**    Table 10 presents the results on the CodeSearchNet AdvTest and WebQueryTest datasets. We report the performance of **RoBERTa** [44] and **CodeBERT** [15].

Table 10: Results on the code search task.

| Model | AdvTest | WebQueryTest | Overall |
|---|---|---|---|
| | MRR | F1 | |
| RoBERTa | 18.33 | 57.49 | 37.91 |
| CodeBERT | **27.19** | **58.95** | **43.07** |

**Text-to-code generation**    Table 11 presents the results on the CONCODE test set. **Seq2Seq** [64] is an RNN-based sequence to sequence model. **Seq2Action + MAML** [22] combines a context-aware retrieval model with model-agnostic meta-learning (MAML). **Iyer-Simp + 200 idoms** [35] extracts code idioms and applies idioms-based decoding. We also report the performance of pretrained models, including **GPT-2** [53], **CodeGPT**, and **CodeGPT-adapted**. The CodeBLEU score is used as the overall score.

Table 11: Results on the text-to-code generation task.

| Model | EM | BLEU | CodeBLEU |
|---|---|---|---|
| Seq2Seq | 3.05 | 21.31 | 26.39 |
| Seq2Action+MAML | 10.05 | 24.40 | 29.46 |
| Iyer-Simp+200 idoms | 12.20 | 26.60 | - |
| GPT-2 | 17.35 | 25.37 | 29.69 |
| CodeGPT | 18.25 | 28.69 | 32.71 |
| CodeGPT-adapted | **20.10** | **32.79** | **35.98** |

**Code Summarization**    Table 12 shows the results achieved by different models in code summarization. **Transformer** and **RoBERTa** use the same setting as **CodeBERT**, but the encoder is initialized randomly and by RoBERTa [44], respectively.

Table 12: Results on the code summarization task.

| Model | Ruby | Javascript | Go | Python | Java | PHP | Overall |
|---|---|---|---|---|---|---|---|
| Seq2Seq | 9.64 | 10.21 | 13.98 | 15.93 | 15.09 | 21.08 | 14.32 |
| Transformer | 11.18 | 11.59 | 16.38 | 15.81 | 16.26 | 22.12 | 15.56 |
| RoBERTa | 11.17 | 11.90 | 17.72 | 18.14 | 16.47 | 24.02 | 16.57 |
| CodeBERT-EncDec | **12.16** | **14.90** | **18.07** | **19.06** | **17.65** | **25.16** | **17.83** |

**Documentation translation**  Table 13 shows the results achieved by the models on eight translation directions. **Transformer** is the multilingual translation model [36]. **XLM-R** initializes the encoder of **Transformer** with XLM-R[12].

Table 13: Results on the documentation translation task.

| Model | EN → DA | EN → LA | EN → NO | EN → ZH | DA → EN | LA → EN | NO → EN | ZH → EN | Overall |
|---|---|---|---|---|---|---|---|---|---|
| Transformer | 53.31 | 37.85 | 53.84 | 59.90 | 58.73 | 50.37 | 57.73 | 50.00 | 52.67 |
| XLM-R | **67.09** | **51.92** | **68.00** | **70.60** | **67.02** | **68.30** | **71.84** | **64.47** | **66.16** |

**Overall Results**  We find pretrained models outperform other models in all tasks. And models pretrained on code corpus (i.e. CodeBERT, CodeGPT, and CodeBERT-EncDec) achieve better results than RoBERTa that only learns from natural language. These experimental results demonstrate that pretraining is useful for code intelligence tasks. However, the improvement is quite limited in some tasks like clone detection, defect detection, code repair, etc. A potential direction for further improvement is to incorporate information from code structures such as Abstract Syntax Tree, data flow, control flow, etc.

## 5   Related Work

Benchmark datasets have been playing a central role in the growth of applied AI research. For example, the LibriSpeech [49] and the SQuAD [54] datasets drive the development of data-driven models for automatic speech recognition and reading comprehension of text, respectively. With the growing demand for testing models' generalization ability on a wide range of applications, researchers have created or assembled datasets that cover many tasks. Representative samples of these datasets include ImageNet [13] for computer vision, GLUE [76] and SuperGLUE[77] for natural language understanding, XTREME [29] and XGLUE [43] for cross-lingual natural language processing. To the best of our knowledge, CodeXGLUE is the first diversified benchmark dataset that can be applied to various code intelligence problems.

Many tasks related to machine learning for software engineering [5] have sufficient amount of data to support the development of data-driven methods, but are not covered by CodeXGLUE. We plan to extend to these tasks in the future. For example, the idiom mining task [2, 35] is to extract code idioms, which are syntactic fragments that recur across software projects and serve a single semantic purpose [2]. Bug localization [55, 24, 71] is to point the error location when a program fails tests. The test case generation task [19, 70] is to generate unit test cases automatically. The program synthesis [47, 58, 62, 74, 17, 40, 90] extends the text-to-code generation task aims to generate programs from a specification [21], such as pseudocode, natural language description, and input/output examples.

## 6   Conclusion

With CodeXGLUE, we seek to support the development of models that can be applied to various program understanding and generation problems, with the goal of increasing the productivity of software developers. We encourage researchers to participate in the open challenge to make progress in code intelligence. Moving forward, we are planning to extend CodeXGLUE to more programming languages and downstream tasks while continuing to develop advanced pretrained models by exploring new model structures, introducing new pretraining tasks, using different types of data, and more.

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
