# OpenReview forum: "CodeXGLUE: A Machine Learning Benchmark Dataset for Code Understanding and Generation"
_NeurIPS.cc/2021/Track/Datasets_and_Benchmarks/Round1 — NeurIPS 2021 Datasets and Benchmarks Track (Round 1)_

### Official Review · Reviewer_rTzs · 2021-06-19
**A useful benchmark dataset for code**

**Rating:** 7
**Confidence:** 4

**Strengths:**

This is a relevant contribution that will likely have a strong impact and be used by the community. Benchmark datasets are a useful asset to evaluate new general-purpose systems and models, and while areas like natural language or image processing have their own dataset suites, this was not true (to my knowledge) in the realm of source code.

The presented dataset suite covers a wide range of tasks (with the intention of extending it in the future), and also competitive baselines.

The dataset is publicly accessible under a license that permits computational use.

**Weaknesses:**

The description of some of the datasets could be clearer. As the description paper for the benchmark dataset, this will be probably a go-to resource and first point of contact for researchers that want to work with these benchmarks. As such, I think it should be written (as far as possible) in such a way that the description of each dataset can be understood without being familiar with the original papers or even tasks, which I think is not true in some cases (concrete examples and suggestions are provided in the "Clarity" section below).

Some items in the checklist are in [TODO] status.

**Additional Feedback:**

I confirm that I have read the author rebuttal. It has reaffirmed my positive recommendation for the paper.

The checklist seems to have a contradiction, it says that no license is specified because the code is proprietary, but the github repository does specify a license for the code.

Two typos:

In line 118, the full stop is separated from the previous line.

In line 157, missing space before opening parenthesis.

**Clarity:**

The paper is generally well-written, but some things should be clarified more, as outlined in sections above.

In particular:

In Table 1, rather than marking the tasks that have new datasets in the task column, it would be clearer to mark the datasets directly in the dataset column. With the current version of the table, it is not clear exactly which datasets are new, one has to go to the text.

In the description of BigCloneBench, please clarify the distinction between "code fragments", "clone pairs" (true and false) and "examples". The dataset (after filtering) is claimed to have ~9K fragments but millions of clone pairs and examples, so I suppose a fragment is a larger piece of code and one fragment can contain N clone pairs (which are the examples) but this is not obvious from the description. A benchmark dataset description paper like this should not assume that the reader knows the dataset (I understand that these details will be described in the original dataset papers, but this paper will be the go-to point for researchers that want to evaluate against the benchmark so it should aim to be reasonably self-contained).

It's also worth explaining why F1 is picked as the metric for that task (which I understand is a binary classification task) while accuracy is picked for e.g. defect detection - I suppose is due to class balance/imbalance?

In the description of CodeRepair, it would be helpful to mention what the scope of the intended repairs is - i.e., whether repairs affect single tokens or lines or they could be wider refactorings.

For the documentation translation task, please specify if the sentence pairs in the dataset have been translated by humans or they have some degree of machine translation.

**Correctness:**

The dataset design and experimental design in the paper look sound. I just have a few minor questions about the choice of some accuracy metrics (in the "Clarity" suggestions below).

**Documentation:**

While the paper does not provide a lot of details of this kind for space reasons (being a suite of datasets that cannot devote a lot of space to each dataset), the github repository does provide comprehensive documentation of each dataset.

**Ethics:**

Not an expert of ethical issues around the use of source code, but I do not see any ethical concerns with this paper.

**Relation To Prior Work:**

The discussion of related work is good, but it would make sense to mention SuperGLUE (https://arxiv.org/abs/1905.00537) as another relevant benchmark dataset in the field of natural language processing.

**Summary And Contributions:**

This paper presents a benchmark dataset focused on source code processing (understanding and generation) tasks. The dataset is rather comprehensive, including code-to-code, code-to-text, text-to-code and text-to-text tasks. It includes 8 preexisting datasets that were already used to evaluate some of the tasks, together with 6 newly-introduced datasets. The paper describes each of the tasks and datasets, and also provides strong baseline models based on pretrained language models and the corresponding results.

---

> ### Author Response · Authors · 2021-07-12
> **Author response**
>
> Thank you so much for your valuable review which helps us to improve our paper.
>
> ### Weaknesses
>
> **TODO items in checklist**
>
> Thank you for your reminder. We have replaced [TODO] with [N/A], which should be what we meant.
>
> ### Clarity
>
> **Table 1**
>
> Thank you for your good suggestion. We have marked the datasets directly in the dataset column in the updated paper.
>
> **BigCloneBench description**
>
> Thank you for your good suggestion. We have made a more clear description in the updated paper. In BigCloneBench, each code fragment is a Java method and all possible pairwise combinations of code fragments are clone pairs. We have replaced 'examples' by 'clone pairs' for clarity.
>
> **F1 metric for BigCloneBench**
>
> Since in the clone detection task the number of false clone pairs is much more than that of true clone pairs, we use F1 score as the metric.
>
> **Code repair description**
>
> We add a brief description on how the dataset is collected in the updated paper. The scope of intended repairs is not restricted. All repairs come from bug-fixing commits in GitHub. The repairs with more than 100 atomic AST modification actions between the buggy and the fixed codes are filtered out.
>
> **Documentation translation task**
>
> Microsoft Docs doesn't have an official statement on how they translate documentations. The community can contribute to the documentations in different languages on the website. We did some case studies on this dataset and we believe they also have some degree of machine translation.
>
> ### Relation To Prior Work
> Thank you for the reminder. We have added SuperGLUE in related works.
>
> ### Additional Feedback
> We have made a more clear description in the updated paper that all the datasets are available under a permissive license that allows computational use purposes, such as artificial intelligence, machine learning, and text and data mining. We also change the answer to [YES] of question 4.(b) in the checklist.

---

> > ### Comment · Reviewer_rTzs · 2021-07-12
> > **Thanks for the changes**
> >
> > I think these changes address the issues I raised, and reaffirm my positive recommendation for the paper.

---

### Official Review · Reviewer_tmjF · 2021-07-05
**A comprehensive benchmark dataset for program understanding/generation**

**Rating:** 7
**Confidence:** 3
**Clarity:** The paper is well-written and easy to…

**Strengths:**

Program understanding and generation is an important and interesting topic that has attracted considerable amount of research efforts recently. The biggest contribution of this paper is to put together a large collection of program-related tasks (several of these tasks are introduced by the authors to fill in the missing parts) that cover (almost) all aspects of this area. I believe this benchmark dataset has the potential to become a standard benchmark in further research and will facilitate the development of program understanding and generation research.


**Weaknesses:**

My only major complaint is that there is no single baseline model that can be applied to all tasks, or a joint model trained on all datasets to provide an overall performance (the model can very simple and naïve). The value of a benchmark dataset consisting of many tasks might be greatly compromised if the tasks are only processed separately.

**Additional Feedback:**

Please see the weaknesses section.

**Correctness:**

The design and implementation of the benchmark is appropriate. An overall metric for evaluating a system on all subtasks simultaneously is not discussed.

**Documentation:**

The benchmark dataset is well documented.

**Relation To Prior Work:**

Most of the related datasets are covered. If space permits, the authors may also include more existing successful models for the tasks in the benchmark.

**Summary And Contributions:**

This paper proposes a comprehensive benchmark dataset for program understanding/generation, consisting of 10 tasks across 14 datasets, where 5 datasets are newly introduced. The collection of these tasks cover code-code, text-code, code-text and text-text settings, providing a complete evaluation platform.

---

> ### Author Response · Authors · 2021-07-11
> **Author response**
>
> Thank you so much for your valuable review which helps us to improve our paper.
>
> ### Weaknesses
>
> **There is no single baseline model that can be applied to all tasks**
>
> Thank you for the good suggestion. Since there are some gaps between different tasks, e.g., the input/output format of classification/generation/retrieval tasks, we choose to use three baseline models (CodeBERT, CodeGPT and CodeBERT with a decoder) for all tasks. Next, we will take it into consideration that to train a unified model for all tasks.
>
> ### Correctness
>
> **Overall metric**
>
> Following the popular benchmarks like GLUE and SuperGLUE in NLP, we plan to use the average scores of all tasks as the overall score.

---

> > ### Comment · Reviewer_tmjF · 2021-07-15
> > **Thanks for the response**
> >
> > Thanks for the response. I am keeping my original rating.

---

### Official Review · Reviewer_7y8Y · 2021-07-05
**Important topic, good execution, but the paper is missing the bigger picture, and how the tasks combine together to solve a real-world problem.**

**Rating:** 7
**Confidence:** 3

**Strengths:**

1. It is a relevant and up-date challenge for language models.
2. The choice of baselines is satisfying.
3. The experimental section is well designed and strikes the right balance between providing as much information as needed, and being easy to digest.
4. The website and hosting platform are carefully designed.
5. I like the specific information about the training and inference costs in the Appendix.
6. The novel datasets are described and released publicly.

**Weaknesses:**

## Major
### 1. The motivation behind the choice of tasks.
The most important missing puzzle is the lack of proper motivation and desiderata that would support the choice of tasks. Measuring the quality of systems aimed to deliver code intelligence demands a proper statement of the problem, and approaching the benchmark in a more top-down manner, where the tasks are chosen on purpose to serve a specific need that is explicitly stated and supported. I would expect a discussion on what steps the developer needs to take in order to come up with the working code, and how the specific tasks are helping him in this regard.  It would be perfectly OK if the authors state that the code-writing process consists of X steps, and here we only took some of them, and here is why. Unfortunately, the paper approaches the benchmarking from the bottom, by selecting some code-related tasks (based on the unspecific terms such as 'popularity', 'reputation', or 'widespread adoption') and then clustering them into similar buckets, based on their features, but leaving the reviewer with the feeling that the scientific community may confuse itself in the longterm when choosing this suite as the main target for future development. I would like to avoid inconsistencies between these chosen tasks and the downstream applications. The paper is not convincing in this regard, and I am not going to recommend it to the community in its current version, but I believe the future is bright for this benchmark.\
Questions: Could you provide some rationale behind the choice of the tasks, and compare the chosen tasks with the ones that were not included?  How are these tasks relevant to the process of software development?
I am willing to substantially increase the score if this aspect will be improved.

### 2. Cloze test tasks
It is unclear how cloze test tasks (when the model has to choose from the limited vocabulary or 2 tokens), would solve real-life problems. These tasks seem somewhat artificial and not grounded in the experience of a programmer.
Are there specific situations, in which the developer is supposed to choose from the limited vocabulary, or is it rather a toy task? What is it aimed to measure? At the moment it demands adding the extra classification heads to the Encoder models, but it seems it may be easily simplified and generalized to the 'next token prediction' task. Is there any argument against that?

### 3. Human baselines are missing
In order to measure progress in the development of code-helping language models, the paper should discuss how well humans perform on these tasks.  Is it possible to perform human evaluation and take into account the experience of a software developer? I believe knowing where we stand with relation to the human baseline should not only be provided, but also be the main criterion when choosing tasks.

## Minor
### 4. Confidence Interval are not provided
As it is stated in the reproducibility checklist, the models were run 2-3 times. The paper would greatly benefit from this information.

### 5. The novelty of some datasets is questionable
The caption in Table 1. states that the 'Code completion' task contains a novel dataset, however, it is not clear what is novelty there, as, in Section 2.4 (Code completion), it is written that the previously developed datasets are used. From the provided description it is hard to understand where the novelty is. Does it lie in the reformulation of the original dataset, or in the performed processing steps that slightly differ from the originally devised?

**Additional Feedback:**

## Questions of minor importance
### 1. Programming languages chosen
What was the driven force behind the choice of programming languages for tasks? Is it data availability or is there a different rationale? How does it stand with the most popular used languages at the moment?

### 2. Aggregating the results and future expansion
It is not described how the results are aggregated in the 'Overall' leaderboard. My concern is that the future extension of the benchmark will change the balance between the tasks and groups of tasks(e.g., code-code). How do you plan to expand the benchmark? Is it going to be backward-compatible?

### 3. Tutorials on how to run the baselines
As the code is provided(which is great!) it may be very helpful to deliver a simple readme on how to reproduce results from the paper and perform inference with the already hosted models.

**Clarity:**

The writing is very good.\
Only minor flaws or typos can be spotted:\
1. line #49 'test test sets' -> 'test sets'\
2. In Table 1: Two tasks lack numbers in the size column, and it should be explained how to understand it.\
3. #118 - dot on the next line
4. #153  - 'AdvTest ' - It is unclear what is the full name. Is it Adversarial, Advanced, or something else?
5. Related Works - I think it is perfectly OK to drop the first paragraph as the work does not benefit from it.

**Correctness:**

The technical side of the evaluation and experimental section is correct. Some issues with the dataset's novelty were raised in 'Weaknesses'

**Documentation:**

Important information is available in Appendices/code.

**Ethics:**

I do not think any ethical concerns should be raised.

**Relation To Prior Work:**

Yes, the work is novel in its narrow field, and a wider comparison to other benchmarks is also provided.

**Summary And Contributions:**

The paper proposes a benchmark to measure the quality of the systems in the growing domain of code intelligence that includes code search, code completion, and code-to-code translations among others. The authors presented three Transformer-based systems (with BERT-style encoder, GPT-style decoder, and encoder-decoder) to set up baselines. Some of the tasks are novel or improve over the original creation process.

=========EDIT=========\
After the discussion phase, the rating was increased from the initial 4 to 7.\
======================

---

> ### Author Response · Authors · 2021-07-11
> **Author response (Part 1)**
>
> Thank you so much for your valuable review which helps us to improve our paper.
>
> ### Weaknesses
>
> **The motivation behind the choice of tasks**
>
> The area of code intelligence lacks a benchmark dataset that covers a wide range of tasks. We choose tasks from code-related tasks and aim to covering the most common scenarios for software development. Assuming there is a developer who wants to build a software system. Since it is difficult to build from scratch, he would like to start with an open-source project to get familiar with how to build a software system. But the developer still finds it difficult to understand the intention of some methods written by others because no documentation is provided. So he needs code summarization systems to generate comments for source codes. In another case, if the documentation is written in a language he can't understand, he can use documentation translation systems to translate the comments. When the developer starts writing codes, he might be confused about what to write next. Code recommendation systems can give him suggestions on the next token, the unfinished line or even a whole method and improve the efficiency of the development process, which is the code completion task and code generation task's purpose. From time to time, he would be stuck by implementing a specific function, e.g., the quick sort algorithm. What he is most likely to do next is to search "how to implement quick sort algorithm" in a search engine and expect to find a solution. This is what the natural language code search task aims to doing. When the developer finds a code snippet which could be helpful to him but he wants to implement the same function in another programming language, he can leverage code translation systems to translate one function from one programming language to another. When he finally finishes building a software system, there might be some bugs or defects in his codes. The defect detection task and the code repair task are to find the bugs and defects and auto-fix them. During the process described above, the developer reuses code fragments from other projects, which leads the software system containing clone codes. Studies have shown that a software system usually contain 7-23\% clone codes. Detecting and refactoring cloned codes are beneficial for maintaining the software quality. That's why we add the clone detection task.
>
> We have added this part to the updated paper.
>
> **Cloze test tasks**
>
> The cloze test task is served as a probing task, which has been widely used in NLP. It does have less contribution to real-life problems compared with other tasks. But it is a nice task for intuitively evaluating models’ ability to understand code semantics. We use limited vocabulary because many of the tokens in codes like punctuations could be useless or semantically irrelevant which might be easier to predict compared with other tokens like *max* and *min*. Besides, developers don't always write codes from left to right. They often modify the parts that have already been written. That's is also why we use the cloze test task but not 'next token prediction' task.
>
> **Human baselines are missing**
>
> Thank you for the good suggestion. We are considering to add human performance for all the tasks. For some tasks, like code completion, code repair, code translation, code summarization and text-to-code generation, these datasets are collected from source codes in real world. So the ground truth comes from human already. On the other hand, for other existing datasets, we will sample and let human to answer.
>
> **Confidence Interval are not provided**
>
> We run our models in each tasks 2-3 times and report the average results. We will make it clear in the final version of our paper.
>
> **The novelty of some datasets is questionable**
>
> We newly introduce the task – line-level completion task, as a subtask of code completion. And we create a new test dataset from the original dataset for it.

---

> > ### Comment · Reviewer_7y8Y · 2021-07-14
> > **Final word from the reviewer**
> >
> > Thank you for the answer. I read the recent revision of the paper and acknowledge the authors did a decent job in increasing the paper clarity(regarding contribution), motivation(regarding the choice of tasks), structure(regarding the narration), and improving the overall self-containdness of a manuscript. In addition, the author's answer and rationale behind the cloze test task are satisfactory.
> > Although some of the issues are yet to be resolved(e.g., adding information about confidence intervals), I believe they are of minor importance and may be easily added to the camera-ready version.
> >
> > As of now, the contributions are solid, and the potential impact seems high. Following this, in the light of new factors and improvements conducted by the authors, I reassessed my valuation of the paper and now consider it to be strong 7.
> >
> > Last typos spotted:
> > - Table 1 caption: introcude -> introduce
> > - `pre-train` vs.`pretrain` -> in different parts of the text, a different way of spelling this word is employed. Frankly, I think it is OK to use `pretrain` consistently.

---

> > > ### Author Response · Authors · 2021-07-15
> > > **Thank you**
> > >
> > > We thank again for your valuable review and your recognition of our work.  We will provide confidence intervals and fix repos in the final version of our paper.

---

> ### Author Response · Authors · 2021-07-11
> **Author response (Part 2)**
>
> ### Additional Feedback
>
> **Programming languages chosen**
>
> We want to focus on the most popular languages and also take the diversity of programming languages into consideration. To some extent, the popularity of a dataset is related to the popularity of the language included in the dataset. So when we use popularity as a criterion, we are choosing the most popular programming language meanwhile. According to PYPL Index, the most two popular programming languages in 2020 are Python and Java. In CodeXGLUE, 11 out of 14 datasets contains Java or Python source codes, As for the diversity, 9 programming language are included in CodeXGLUE in total.
>
> **Aggregating the results and future expansion**
>
> Following the popular benchmarks like GLUE and SuperGLUE in NLP, we plan to use the average scores of all tasks as the overall score.
>
> **Tutorials on how to run the baselines**
>
> We provide the readme in our GitHub page to reproduce the experiments, including training and inference, for all tasks. E.g., [here](https://github.com/microsoft/CodeXGLUE/tree/main/Code-Code/Clone-detection-BigCloneBench#pipeline-codebert) is the instruction on how to fine-tune CodeBERT model on the clone detection task.

---

### Author Response · Authors · 2021-07-12
**General Response: Summary of Updated Paper**

We thank all reviewers for their valuable reviews and suggestions. We have improved our paper based on their thoughtful suggestions. The key changes are as below.

1. We clarity our motivation on the choice of tasks and how these tasks help real-life software development in the beginning of Section 2 according to Reviewer 1's suggestion.
2. We mark the new datasets directly in the dataset column in Table 1 for clarity suggested by review 3.
3. We make more clear descriptions of the tasks according to reviewer 1's and reviewer 3's suggestions, including Section 2.1, 2.5 and 2.10.
4. Fix typos based on reviewers' comments.

---

### Decision · Program_Chairs · 2021-07-26

**Decision:**

Accept

**Comment:**

The paper proposes a comprehensive benchmark for program understanding and generation consisting of 10 tasks across 14 datasets. All the reviewers agree that this dataset has potentially be used as a standard benchmark to measure progress in this domain. Therefore, I recommend acceptance of this paper.